# The Usefulness of Peroral Cholangioscopy for Intrahepatic Stones

**DOI:** 10.3390/jcm11216425

**Published:** 2022-10-29

**Authors:** Yuri Sakamoto, Yohei Takeda, Yuta Seki, Shiho Kawahara, Takuya Shimosaka, Wataru Hamamoto, Hiroki Koda, Taro Yamashita, Takumi Onoyama, Kazuya Matsumoto, Kazuo Yashima, Hajime Isomoto

**Affiliations:** Division of Gastroenterology and Nephrology, Department of Multidisciplinary Internal Medicine, Faculty of Medicine, Tottori University, Nishi-cho 36-1, Yonago 683-8504, Japan

**Keywords:** endoscopic retrograde cholangiopancreatography, electrohydraulic shockwave lithotripsy, gallstone, stenosis, transpapillary treatment

## Abstract

Peroral cholangioscopy (POCS) is believed to be effective in treating intrahepatic stones; however, reports on its efficacy are few. We reviewed the results of intrahepatic stones treated with fluoroscopic guidance or POCS. This study included 26 patients who underwent endoscopic treatment for intrahepatic stones at our institution between January 2017 and December 2021. We retrospectively evaluated the procedure time and adverse events in the first session and the rate of complete stone removal. Complete stone removal was achieved in 92% (24/26); POCS was required in 16 of 26 (62%) procedures and the complete stone removal was achieved in 15 of 16 (94%) of these procedures. The POCS group had a significantly longer procedure time than the fluoroscopy group. Cholangitis incidence was high; however, no difference was noted between patients with and without POCS, and all cases were mild and treated conservatively. Endoscopic treatment for intrahepatic stones may lead to an increase in the incidence of cholangitis, requires specialized devices such as a cholangioscope, and should be performed in an established institution by experienced staff. POCS is useful for intrahepatic stones formed upstream of the stenosis and intrahepatic stones piled in the bile duct.

## 1. Introduction

Non-surgical treatment for intrahepatic stones is the mainstream alternative to surgical treatment in Japan [1]. In addition, the frequency of transpapillary treatment has increased, replacing percutaneous transhepatic cholangioscopic lithotripsy (PTCS) and extracorporeal shockwave lithotripsy (ESWL).

Balloon or basket catheters are the first choices for transpapillary gallstone treatment [2]. Endoscopic papillary large balloon dilation (EPLBD) and endoscopic mechanical lithotripsy (EML) are useful for large stones, and electrohydraulic shockwave lithotripsy (EHL) and laser lithotripsy (LL) are effective when EML fails to crush the stones. Reports of endoscopic treatment with peroral cholangioscopy (POCS) have increased since the launch of the disposable cholangioscope, SpyGlass^TM^DS.

Some intrahepatic stones are caused by congenital or acquired anatomic abnormalities, stenosis, or deformity of the bile ducts and are often difficult to treat endoscopically under fluoroscopic guidance; cholangioscopy may be useful in their treatment [3,4]. This study examined the results of transpapillary treatment of intrahepatic stones at our hospital.

## 2. Materials and Methods

### 2.1. Patients

This single-center retrospective study was performed at Tottori University Hospital. Data were obtained for all patients aged 20 years and older. Twenty-six patients who underwent endoscopic treatment for intrahepatic stones between January 2017 and December 2021 were enrolled. This study was approved by Ethics Committee of Tottori University Hospital.

### 2.2. Materials and Method

Prophylactic antibiotics were administered in 18 cases treated after February 2019. A duodenoscope (TJF-Q290V or JF-260V; Olympus Medical Science Corporation, Tokyo, Japan) was used for patients with normal anatomy, and a double-balloon endoscope (EI-580BT; FUJI FILM Corporation, Tokyo, Japan) or colonoscope (CF-HQ290ZI; Olympus Medical Science Corporation, Tokyo, Japan) was used for patients with postoperative anatomy. When necessary, scopes were changed during the procedure.

Endoscopic sphincterotomy (EST) was performed for transpapillary procedures; endoscopic papillary balloon dilation (EPBD) or EPLBD was added in post-EST patients when necessary. For patients who underwent hepatico-jejunostomy, balloon dilation was performed if stenosis of the anastomosis was observed.

If there was stenosis in the bile duct, balloon dilation or stenting was performed. Stone removal was initiated with fluoroscopic guidance using a basket or balloon catheter and then switched to POCS with a SpyGlass^TM^DS (Boston Scientific Corporation, Marlborough, MA, USA) as needed. Two hours after the start of the procedure, a decision was made whether to stop or continue treatment.

### 2.3. Case 1: EHL under POCS with Normal Anatomy

The patient was admitted for treatment of cholangitis due to a right intrahepatic stone. TJF-Q290V was advanced to the major duodenal papilla. A guidewire (M-Through, ASAHI INTECC CO., Ltd., Aichi, Japan) and catheter (Glo-Tip II^®^, Cook Medical, Bloomington, IN, USA) were used to cannulate the bile duct. Since the patient had a history of EST, an EPBD (ZARA, Century Medical, Inc., Tokyo, Japan) was added, and a SpyGlass^TM^DS was inserted into the bile duct. The right anterior sectoral duct could not be identified due to stenosis, and the first session ended with observation only. In the second session, the right anterior sectoral duct was identified by SpyGlass, and a guidewire (JagWire^TM^, Boston Scientific Corporation) and a catheter (PR-110Q-1, Olympus Medical Science Corporation) were successfully inserted; the stenosis was dilated with a bile duct dilating balloon (REN^®^, KANEKA CORPORATION, Tokyo, Japan), and a SpyGlass was inserted. We repeated stone removal under fluoroscopic guidance and POCS, and complete stone removal (CSR) was successfully achieved after seven treatments (Figure 1).

### 2.4. Case 2: Treatment with a Metal Stent to Expand Stenosis in a Patient after Right Hepatectomy

The patient had undergone a right hepatectomy for hepatocellular carcinoma. Postoperatively, the left intrahepatic bile duct became stenosed and stones formed upstream of the stenosis. After four sessions of treatment, including EHL under POCS, the stone was almost completely removed. However, the stenosis remained and was difficult to dilate with a plastic stent; hence, a self-expandable metal stent (BONASTENT^®^ M-Intraductal, MEDICO’S HIRATA Inc., Osaka, Japan) was placed. When the stent was removed one month later, the stenosis had expanded (Figure 2).

### 2.5. Case 3: EHL under POCS with Colonoscopy with Postoperative Anatomy

The patient had undergone pancreaticoduodenectomy. Seven years after surgery, the patient developed cholangitis due to intrahepatic stones. Although the hepatico-jejunostomy was reached with a double-balloon endoscope, the stones were piled up, and the fluoroscopic approach was difficult. While withdrawing the double-balloon endoscopy, the afferent loop was marked by endoscopic tattooing with India ink. A colonoscope was inserted along the tattoo to the hepatico-jejunostomy. EHL under POCS was performed by SpyGlass^TM^DS, and CSR was successful (Figure 3).

### 2.6. Outcome Measurements

We retrospectively evaluated the procedure time and adverse events in the first session and the rate of CSR. The procedure time was measured from scope insertion to removal. Adverse events were reviewed per patient and graded according to the American Society for Gastrointestinal Endoscopy lexicon’s severity grading system. CSR was defined as the absence of stones on cholangiography at the end of the procedure. The size and number of stones were measured and counted by cholangiography. Descriptive statistics are shown as medians and range. The Mann–Whitney U test was used to compare the median of continuous variables and the chi square test to compare the proportions of categorical variables between the groups.

## 3. Results

Table 1 shows the characteristics of the patients. Twenty-six patients (18 males and eight females) with a median age of 64.5 (26–79) years were included in this study. Fourteen patients had normal anatomy, and 12 had postoperative anatomy, including eight patients with modified Child surgery and four with bile duct resection for congenital biliary dilatation, gallbladder carcinoma, or recurrent common bile duct stones. Five patients had undergone hepatectomy, four had sclerosing cholangitis (including suspected cases), and three had peribiliary cysts. The median maximum diameter of intrahepatic stones was 6 (3–21) mm, and the number of stones was one (*n* = 2), two (*n* = 2), three (*n* = 1), and four or more (*n* = 22).

Table 2 shows the results of endoscopic treatment. Eleven patients underwent EHL under POCS in the first session; five patients achieved CSR in one session, and five eventually achieved CSR by EHL under POCS (*n* = 3), PTCS (*n* = 1), or fluoroscopic guidance (*n* = 1) at the second to fourth session (Figure 4).

Fifteen patients were treated under fluoroscopic guidance in the first session; nine patients achieved CSR in one session; five eventually achieved CSR with EHL under POCS (*n* = 4) or basket catheter under POCS (*n* = 1) (Figure 4). Therefore, POCS enabled CSR in five of six patients who had failed fluoroscopic guidance treatment.

CSR was achieved in 92% (24/26) of cases; CSR was not achieved in two cases. One case was an intrahepatic stone found incidentally and judged unnecessary to treat. The other case was after a bile duct resection for congenital biliary dilatation, and the double-balloon endoscope could not reach the anastomosis site. The median number of sessions in patients who achieved CSR was one (1–7). In summary, 62% (16/26) patients required POCS, and 94% (15/16) achieved CSR.

The median procedure time for the first session was 97 (27–220) minutes, whereas that of the EHL under POCS group was 121 (43–213) minutes and that of the fluoroscopic guidance group was 77 (27–220) minutes, indicating that the EHL under POCS group had a significantly longer procedure time (*p =* 0.0195, Mann–Whitney U test). Cholangitis developed in six (24%) and pancreatitis in three (12%) cases after the first session; however, all cases were mild and could be treated conservatively including antibiotic treatment. Although the incidence of cholangitis was high, there were no differences in complication rates between patients with and without EHL under POCS (*p =* 1, chi square test).

## 4. Discussion

The term “difficult stones” is a general term for stones that are difficult to treat with standard removal techniques, such as large stones, piled stones, and intrahepatic stones. Lithotripsy under POCS using SpyGlass has improved the results of treating difficult stones [5]. Although it is assumed that SpyGlass is effective in treating intrahepatic stones, there are few reports on its efficacy [6]. Our study may be considered novel because it examined treatment results that focused on intrahepatic stones.

Before the advent of SpyGlass, the complete stone removal rate for intrahepatic stones by POCS was 64% [7]. In reports of POCS for common bile duct stones and not limited to intrahepatic stones, the complete removal rate of stones was 82–95% [8,9]. In our study, CSR was achieved in 94% (15/16) of cases in which POCS was performed, which is considered a good result. We consider that POCS may play a critical role in the treatment of intrahepatic stones, because 62% (16/26) of cases required treatment under POCS. We propose there are two types of cases in which POCS is useful: (1) intrahepatic stones formed upstream of the stenosis and (2) intrahepatic stones piled up in the bile duct. When there is significant stenosis in the bile duct, as shown in Figure 1, cholangiography may not reveal the area upstream of the stenosis, or it may be challenging to pass a guidewire or catheter through the stenosis. In several cases, the POCS allows the guidewire to be passed through the stenosis with direct vision, and the cholangioscope can be pushed through the stenosis while keeping the axis. When the bile duct is piled up with stones, as shown in Figure 3, a guidewire or catheter may not enter the duct, and a basket catheter may not spread. Performing lithotripsy under POCS creates space in the bile duct and allows the removal of stones with a basket or balloon catheter. However, as a note of caution, performing POCS significantly prolongs the procedure time, as was seen in this study.

The incidence of cholangitis is 0.6–1.4% in reports of standard ERCP [10,11], but increases to 6–7.5% in reports of POCS [8,12]. There is a report of a higher rate of post-procedure cholangitis compared to the standard ERCP because POCS requires saline irrigation [13]. In this study, the incidence of cholangitis was high at 24%; however, no difference was observed in incidence between patients with and without POCS. We suspect that cholangitis was caused not only by irrigation of POCS but also by bile stasis due to edema of the bile duct caused by catheter manipulation because of the narrow intrahepatic bile duct. To prove this, it would be necessary to compare the incidence of cholangitis with the treatment results for common bile duct stones.

The incidence of pancreatitis was 12% in this study. This is higher than the previously reported incidence of pancreatitis in POCS (7–8.9%) [8,12], but the small number of cases should be considered. In addition, in this study, two of the three cases of pancreatitis were observed in patients who had undergone hepaticojejunostomy and were treated with double-balloon endoscopy.

In conclusion, endoscopic treatment for intrahepatic stones should be performed at established centers, because specialized devices such as a cholangioscope may be required and cholangitis may increase. POCS may increase the procedure time, but it is useful in the treatment of intrahepatic stones.

## Figures and Tables

**Figure 1 jcm-11-06425-f001:**
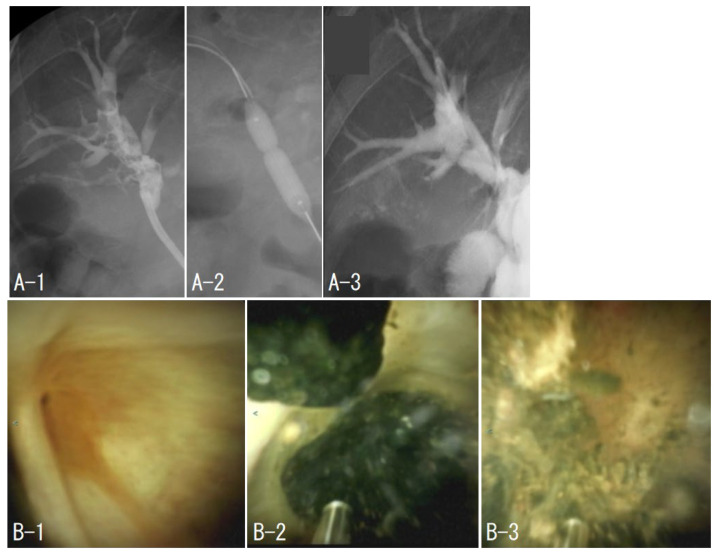
Case 1. The beginning of the right anterior columnar canal was narrowed (**B-1**), and stones filled the upstream of the stenosis (**A-1**). Balloon dilation of the stenosis was performed (**A-2**). The stone was removed by electrohydraulic shockwave lithotripsy under peroral cholangioscopy (**B-2**,**B-3**), and cholangiography was performed to confirm the absence of residual stones (**A-3**).

**Figure 2 jcm-11-06425-f002:**
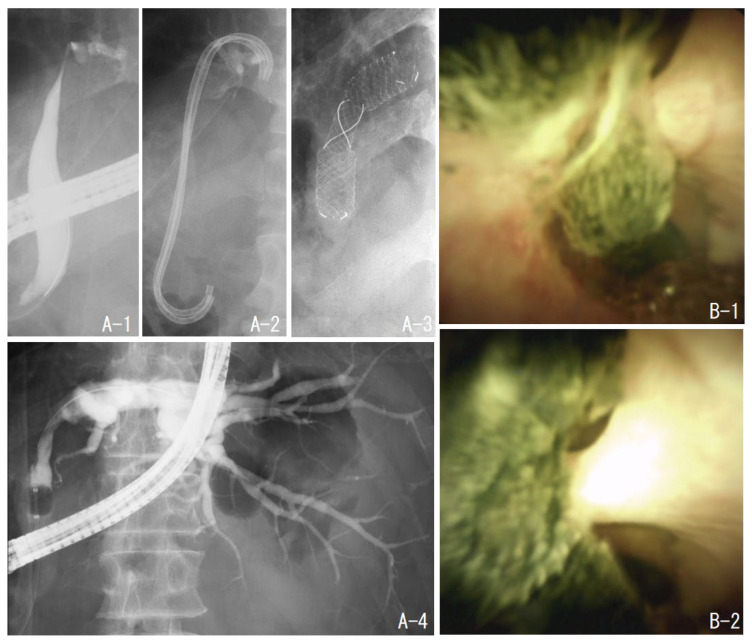
Case 2. The patient had a stenosis of the left hepatic duct after the right hepatectomy (**A-1**). There were multiple stones upstream of the stenosis (**B-1**,**B-2**). The stones were removed by electrohydraulic shockwave lithotripsy under oral cholangioscopy. However, the stenosis was not dilated by the plastic stent (**A-2**); hence, a self-expandable metallic stent was placed (**A-3**). One month later, the stent was removed, and cholangiography showed improvement of the stenosis (**A-4**).

**Figure 3 jcm-11-06425-f003:**
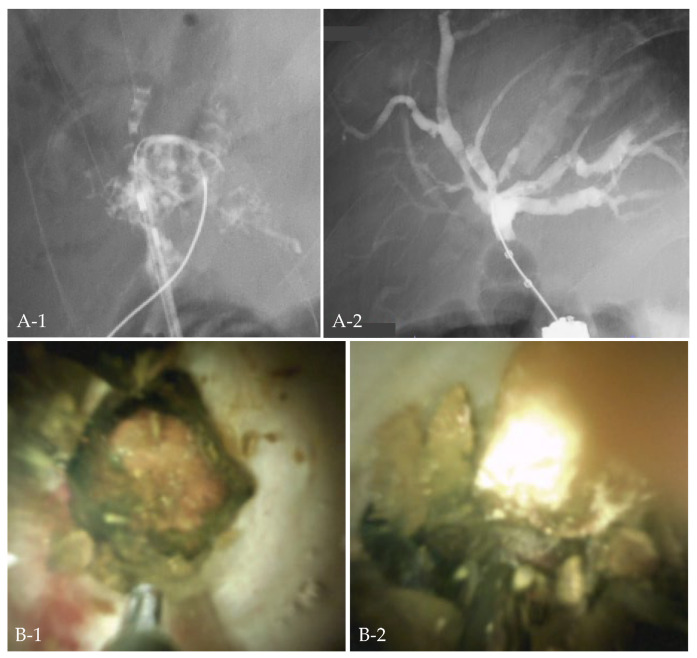
Case 3. Cholangitis developed in a patient after pancreaticoduodenectomy, and the bile duct was filled with stones (**A-1**). Electrohydraulic shockwave lithotripsy under oral cholangioscopy with colonoscopy was performed (**B-1**,**B-2**), and all stones were removed (**A-2**).

**Figure 4 jcm-11-06425-f004:**
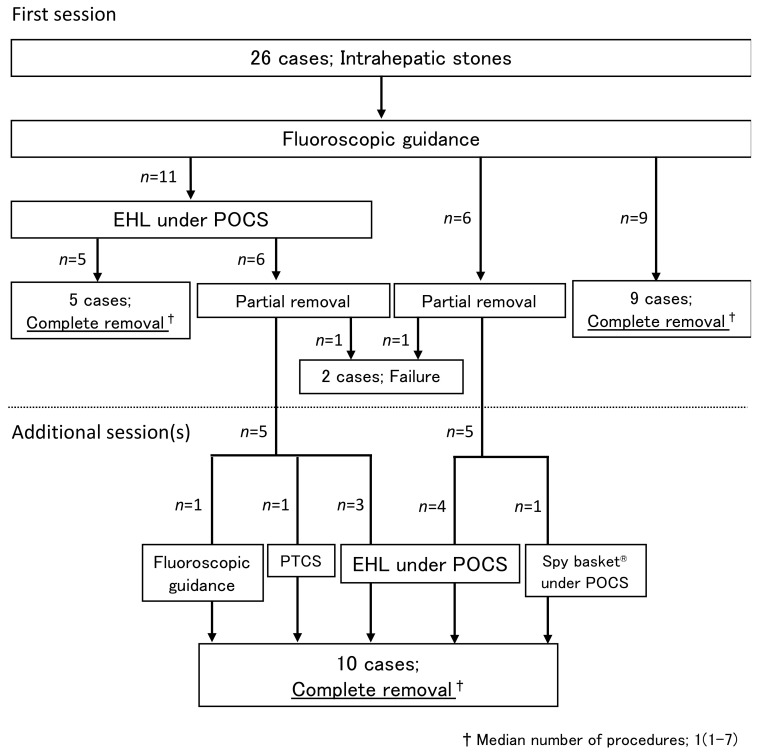
Progression of treatment. Abbreviations: EHL, electrohydraulic lithotripsy; POCS, peroral cholangioscopy; PTCS, percutaneous transhepatic cholangioscopic lithotripsy.

**Table 1 jcm-11-06425-t001:** Patient characteristics.

	*n* = 26
Clinical features	
Age (median)	64.5 (28–79)
Male/female	18/8
Previous surgery	
Modified Child method	9 (34%)
Flow diversion operation	3 (12%)
Hepatectomy	5 (19%)
Coexisting disease	
Sclerosing cholangitis	5 (20%)
Peribiliary cysts	3 (12%)
Stone	
Size (mm) (median)	6 (3–21)
Number	
1	2
2	1
3	1
≥4	22

**Table 2 jcm-11-06425-t002:** Results of endoscopic treatment.

	*n* = 26
Complete stone removal (first session)	14 (54%)
Complete stone removal (last session)	24 (92%)
Procedure time (minutes) (median)	97 (27–220)
Complications	
Pancreatitis	3 (12%)
Cholangitis	6 (24%)

## Data Availability

The datasets generated and/or analyzed during the current study are available from the corresponding author on reasonable request.

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
