# Peer review of "The Usefulness of Peroral Cholangioscopy for Intrahepatic Stones"

_jcm, 2022, doi:10.3390/jcm11216425_

Round 1
Reviewer 1 Report
The authors have retrospectively studied patients who underwent endoscopic treatment for intrahepatic stones over a four-year period at one center. The manuscript contains an introduction, and in the Methods section there are three case reports followed by a Results section that presents overall numbers, figures and tables.
General comments:
The major parts of the three case reports in the methods section seem to fit better in the Results section, although some of it belong in the methods section. The pictures could still be used to illustrate the procedures as well as the stones and stenoses.
It is unclear who patients were selected for POCS. In line 59 it is stated that procedures started under fluoroscopic guidance but in Figure 4 it seems that 11 patients went directly to EHL under POCS? The one patient treated with Spy basket under POCS seems to belong in the group with POCS, the point is not the use of fluoroscopy or not – is it? In conclusion, Figure 4 is difficult to understand and should be simplified in some way.
In the Methods section you describe that you used t-test and chi square tests, but such analyses miss in the Results section. If you planned to divide patients in those needing POCS and not, that would be interesting and comparisons could also be done.
Was prophylactic antibiotics administered before, during or immediately after all POCS procedures – which is commonly performed at many centers? The number of procedures per patient varied a lot. Are complications presented per patient of per procedure? Please comment and specify in the manuscript.
Discussion: There should be a more extensive comparison of the current findings with the preexisting literature. The manuscript contains only eight references. Although few papers may focus only on intrahepatic stones removed by POCS , there must be some numbers to compare your results with found in other studies. The need for comparisons also goes for the cholangitis rate and pancreatitis rate.
Minor comments:
Abstract: the statement that POCS allowed complete stone removal in 58% seems misleading in this context. Please rephrase lines 20-21 to for instance:. POCS was required in 16 of 26 (62%) procedures and complete stone removal was achieves in 15 of 16 (94%) of these procedures.
Line 78: there as apparently not an upper limit of 2 hours for all procedures? Please state such information in the methods section rather than in the case reports.
Line 77. Please specify that REN is a biliary dilation catheter.
Line 82: What is meant by a “strong stenosis”? A fibrotic stenosis or a narrow stenosis?
Line 111. “A patient occurred cholangitis” should be rephrased. For instance “Cholangitis developed in a patient after…”
Table 1: move headlines left instead of centering all text.
Lien 168: “Cholangitis was present” should be rephased to “Cholangitis developed”. If antibiotic treatment was included in “conservative”, please state this.
Figure 4. Abbreviations used in the figure should be explained in the figure legend.
Reviewer 2 Report
This is an interesting paper describing the usefulness of endoscopic treatment for intrahepatic stones specifically focusing on the advantages of peroral cholangioscopy improved since the launch of SpyGlass (TM)DS
It represents a novelty in treating both patients with normal or post-operative anatomy with challenging intrahepatic lithiasis.
I suppose this is a good quality paper even if some amelioration should be made in english language and statistical analysis.
For example in terms of english spelling:
row 18-19 "the" rate and "the complete stone removal"
row 25 "It requireS" and “be performed IN”
Moreover, I think it would be valuable to add a subheading before row 114 since the content explicated is different from the previous “EHL under POCS with colonoscopy with post-operative anatomy”
Considering the amount of patients analyzed, a modification in statistical analysis is needed. Indeed, a 26-patients cohort is too small for applying the means concept as well as t-test. It should be preferred the median with mann-whitney test.
Row 88: The term “right lobectomy” must be revised, since it may create confusion and the Brisbane nomenclature should be preferred.
It would be interesting to understand if the 2h limit to interrupt the procedure is a standard or it was performed only in some cases.
The authors usually emplyed the terms “high” referring to incidence of cholangitis or “good result” when speaking about percentage of CSR achievement. It would be more appropriate to cite previous experiences reported in literature.
Round 2
Reviewer 1 Report
The authors have added information to the revised manuscript and restructured the contents. Results from additional six related publications have been referred to in the Discussion and some information about antibiotic prophylaxis has been added.
Please proof-read the manuscript carefully for minor errors.
Linge 52 Headline: Patients (in plural)
Line 115: Please replace “with colonoscopy…..” with “with a colonoscope in a patient with postoperative anatomy”
Reviewer 2 Report
The authors ameliorated their paper increasing its value by making more fluid the contents explicated. I really appreciate the work they did believing that at this point, the paper is valuable to be published.